# Autophagy and Intracellular Membrane Trafficking Subversion by Pathogenic *Yersinia* Species

**DOI:** 10.3390/biom10121637

**Published:** 2020-12-04

**Authors:** Marion Lemarignier, Javier Pizarro-Cerdá

**Affiliations:** 1Yersinia Research Unit, Institut Pasteur, F-75015 Paris, France; marion.lemarignier@pasteur.fr; 2Université de Paris, Sorbonne Paris Cité, F-75013 Paris, France; 3‘Plague Maintenance, Spread and Evolution’ Pasteur International Unit, F-75015 Paris, France; 4‘Plague and Other Yersinioses’ National Reference Laboratory, F-75015 Paris, France; 5WHO Collaborative Centre for Plague FRA-140, F-75015 Paris, France

**Keywords:** *Yersinia enterocolitica*, *Y. pseudotuberculosis*, *Y. pestis*, *Y. ruckeri*, autophagy, enteric yersiniosis, plague

## Abstract

*Yersinia pseudotuberculosis*, *Y. enterocolitica* and *Y. pestis* are pathogenic bacteria capable of causing disease in humans by growing extracellularly in lymph nodes and during systemic infections. While the capacity of these bacteria to invade, replicate, and survive within host cells has been known for long, it is only in recent years that their intracellular stages have been explored in more detail. Current evidence suggests that pathogenic *Yersinia* are capable of activating autophagy in both phagocytic and epithelial cells, subverting autophagosome formation to create a niche supporting bacterial intracellular replication. In this review, we discuss recent results opening novel perspectives to the understanding of intimate host-pathogens interactions taking place during enteric yersiniosis and plague.

## 1. Introduction

The genus *Yersinia* includes several pathogenic bacterial species for humans, which include the food-borne enteropathogens *Y. enterocolitica* and *Y. pseudotuberculosis* [1], and the vector-borne pathogen *Y. pestis*, the etiologic agent of plague [2]. During *Y. enterocolitica* and *Y. pseudotuberculosis* host infection, interaction between the bacterial surface molecule invasin and host β1-integrins promotes bacterial internalization into intestinal tract cells and colonization of Peyer’s patches and the cecum [3,4,5]. Traversal of the intestinal barrier leads to bacterial draining by local mesenteric lymph nodes, where bacterial extracellular proliferation takes place [6]. Invasin is inactivated in *Y. pestis* [7] and inoculation through flea bites favors pathogen draining by inguinal, axillary or cervical lymph nodes, where bacterial proliferation takes also place extracellularly [8]. Inhibition of phagocytosis and bacterial extracellular life are promoted by *Yersinia* outer proteins (Yops), translocated into host cells by a type 3 secretion system (T3SS) encoded in the *Yersinia* virulence plasmid (pYV in *Y. enterocolitica* and *Y. pseudotuberculosis*, pCD1 in *Y. pestis*) [9,10].

The early signaling cascades that trigger *Y. enterocolitica* and *Y. pseudotuberculosis* entry within epithelial cells upon invasin/β1-integrins interaction have been well studied [11,12,13]. The extracellular stages that characterize lymph node infection by all pathogenic *Yersinia* have been also well described in the literature [14]. However, the bacterial intracellular stages have been less investigated, despite their importance for early infection or bacterial persistence in the intestinal barrier during enteric yersiniosis [3,4,5]. Replication within the intracellular environment of phagocytic cells seems also to play a major role for pathogenic *Yersinia* virulence in vivo [15,16]. Nevertheless, the precise nature of the intracellular compartments supporting bacterial replication remained unknown.

In the last eleven years, different studies have highlighted that *Y. enterocolitica*, *Y. pseudotuberculosis*, and *Y. pestis* can promote bacterial intracellular survival by subverting autophagy, a conserved eukaryotic cascade involved in cellular recycling of macromolecules and organelles, but also in pathogen elimination [17]. In this article, we summarize and discuss relevant articles describing how pathogenic *Yersinia* interfere with membrane trafficking and favor bacterial replication during intracellular infection stages by activating autophagy.

## 2. Autophagy

Autophagy is an essential cellular homeostatic process highly conserved in eukaryotes. It directs cytoplasmic cellular components (damaged organelles, misfolded or aggregated proteins) to endo-lysosomal compartments for degradation and recycling. The pan-eukaryotic distribution of core components of the autophagic machinery argues for the presence of autophagy in the last common eukaryotic ancestor [18]. Autophagy probably played a major role during early eukaryotic evolution by allowing cells to survive under starvation conditions and to localize new foraging locations while consuming intracellular pools of amino acids. Currently, autophagy plays major roles in embryogenesis, placentation, metabolism, cardiovascular health, and neural development. Consequently, autophagy dysfunctions contribute to various pathological processes such as tumor progression, neurodegeneration, or heart failure [19,20]. Autophagy probably played also a major role in the evolution of the immune system, and it is now important for inflammasome activation, type I interferon production, antigen presentation, and pathogen degradation [21].

Three major forms of autophagy are described based on their mode of cargo delivery to lysosomes: (a) chaperon-mediated autophagy (CMA) involves chaperon recognition of cytosolic proteins that are directly translocated across the lysosomal membrane for degradation; (b) microautophagy consists of direct invagination of cytoplasm within a lysosome, and (c) macroautophagy involves isolation of cytoplasmic particles or organelles within a double-membraned compartment that will subsequently fuse with lysosomes [22]. Pathogens can be selectively targeted for degradation by macroautophagy (called in this specific case xenophagy). LC3-associated phagocytosis (LAP) is a non-canonical autophagic pathway that also contributes to pathogen elimination. Macroautophagy/xenophagy and LAP share molecular actors such as evolutionary conserved autophagy-related (ATG) genes and phosphatidylinositol 3-kinase class III (PIK3C3), regulating the different autophagic steps [23].

Macroautophagy/xenophagy is initiated through the kinase AMP-dependent activation of the ULK1 serine threonine kinase complex (ULK1, FIP200, ATG13, ATG101). The activated ULK1 complex translocates to an isolated cellular membrane and becomes the phagophore nucleation site, where it phosphorylates multiple effectors (including itself) [24]. Within the targeted membrane, the autophagy-specific PIK3C3 synthesizes phosphatidylinositol-3-phosphate (PI3P) allowing the recruitment of specific effectors from the WIPI protein family, which participates in phagophore biosynthesis by recruiting ATG effectors [25]. The phagophore elongates thanks to two main ubiquitin-like systems: the Atg5-Atg12 conjugation system, and the microtubule-associated protein light chain 3 system (LC3, the mammalian ortholog of the yeast protein Atg8) [26]. The elongated phagophore curves and fuses between its two ends, forming a double membrane compartment named autophagosome which encapsulates cytoplasmic organelles (or pathogens in the case of xenophagy). The autophagosome matures by fusing directly with lysosomes, building an acidic and degradative autophagolysosome or autolysosome. Before fusing with lysosomes, the autophagosome can fuse with endocytic compartments such as early and late endosomes, as well as multivesicular bodies, forming an intermediary compartment named amphisome [27].

LAP is directly activated by extracellular particles (pathogens or dead cells) binding to pathogen recognition receptors [28], receptors that detect phosphatidylserine (TIM4) [29], or antibodies (FcγR2a) [30]. As it happens in xenophagy, synthesis of PI3P by the PIK3C3 occurs but directly at the phagosome site. Both ubiquitin-like conjugation systems Atg5-Atg12 and Atg8/LC3 are also implicated in the LAP pathway, forming a LC3-positive phagosome called LAPosome, which matures by fusing with lysosomes for degradation [31].

LC3 is used as a hallmark of autophagy in many studies [32]. Indeed, LC3 is involved in different autophagic steps such as: phagophore elongation [33], tethering and membrane fusion [34], interaction with selective autophagy receptors, and autolysosome inner-membrane degradation [35]. The cytosolic form of LC3 is cleaved on its C-terminal part by the cysteine protease Atg4, exposing a glycine residue (LC3-I). The E1-like enzyme Atg7 transfers LC3-I to the E2-like enzyme Atg3. Then, Atg5-Atg12 facilitates the conjugation of the phosphatidylethanolamine to the C-terminal glycine. This allows LC3-II insertion in the inner and outer membrane of the autophagosome, or directly within the single membrane of the LAPosome [22]. LC3-II is then degraded within the autophagolysosome [36].

## 3. *Yersinia pseudotuberculosis*

In vitro analyses have demonstrated that *Y. pseudotuberculosis* can establish an intracellular stage within macrophages by inducing autophagy (Figure 1). Indeed, in murine bone marrow-derived macrophages (BMDMs), Moreau and colleagues (2010) [37] showed that intracellular and metabolically active *Y. pseudotuberculosis* trigger the autophagic machinery in a process independent of the pYV-encoded T3SS. Ultrastructural electron microscopy analyses showed that *Y. pseudotuberculosis* survives and replicates within BMDMs inside autophagosomal compartments formed by double or multiple membranes. Fluorescence microscopy further demonstrated that during the course of infection, *Y. pseudotuberculosis*-containing vacuoles (YCVs) enlarge progressively overtime by fusing with LC3-positive membranes, and harbor the lysosomal associated membrane protein 1 (LAMP1). Upon measuring of the autophagic flux (assessing the conversion of cytosolic LC3 I towards lipidated/membrane-associated LC3 II), as well as quantifying the levels of vacuolar LC3-GFP in *Y. pseudotuberculosis* infected cells, the authors confirm that bacterial infection induces autophagy. Bacterial replication can be detected in *Atg5^−/−^* (autophagic pathway impaired) mouse embryonic fibroblasts (MEFs), but is significantly lower to that observed in wild type MEFs, indicating that autophagy is required for optimal intracellular *Y. pseudotuberculosis* growth. In experiments using BMDMs transfected with the inactive Atg4B C74A mutant of Atg4B (a processing protease for LC3), bacteria are found in acidic compartments labeled with LysoTracker. In wild type BMDMs, conversely, *Y. pseudotuberculosis* is found in non-acidic compartments, indicating that bacteria are able to impair the maturation of YCVs into degradative compartments. This result confirms previous observations indicating that *Y. pseudotuberculosis* inhibits the vacuolar proton ATPase (V-ATPase) [38]. Interestingly, bacteria-free autophagosomes fuse with lysosomes, indicating that the basal autophagic flux is not impaired in infected macrophages [37].

As demonstrated in macrophages, a subsequent study by the same group described that in epithelial HeLa cells, *Y. pseudotuberculosis* also activates autophagy (Figure 2) [39]. In these epithelial cells, *Y. pseudotuberculosis* inhibits the maturation of YCVs, favoring bacterial proliferation in non-acidic compartments. However, and by contrast to what occurs in BMDMs, in HeLa cells, the non-conventional autophagic pathway LAP is engaged with direct recruitment of LC3 proteins to the YCVs. This process triggers the formation of a single-membrane niche for bacterial replication called LAPosome. Fluorescent microscopy experiments then revealed that VAMP3 and VAMP7 are sequentially recruited to the YCVs in both BMDMs and HeLa cells. siRNA inactivation assays demonstrated that VAMP7 facilitates the recruitment of LC3 to YCVs. Interestingly, VAMP3 is found in a higher proportion in YCVs in HeLa cells than in BMDMs. A combination of overexpression and siRNA experiments demonstrated that VAMP3 controls a checkpoint at which bacteria are committed to single LC3-positive compartments [39].

## 4. *Yersinia enterocolitica*

As observed for *Y. pseudotuberculosis*, *Y. enterocolitica* has been also shown to activate autophagy in macrophages and epithelial cells (Figure 1; Figure 2). Deuretzbacher and colleagues (2009) [40] demonstrated that in murine J774A.1 macrophages, *Y. enterocolitica* WA cured from the pYV plasmid triggers the conversion of cytoplasmic LC3 I towards LC3 II. These authors observed intracellular bacteria in multimembrane compartments using electron microscopy as well as bacteria associated to LC3-GFP-positive vacuoles, observed using fluorescence microscopy. The activity of the T3SS, the absence of invasin, or chemical interference with invasin/β1 integrin-mediated signaling, which all lead to reduced bacterial intracellular numbers, reduce autophagy activation and bacterial association with autophagosomes. Interestingly, and opposed to what had been proposed for *Y. pseudotuberculosis* in BMDMs, the authors suggest that autophagy does not favor the creation of an intracellular replication niche for *Y. enterocolitica* in J774A.1 cells, and instead its activation promotes killing of intracellular bacterial. In the same line, chemical inhibition of autophagy or of vacuolar acidification leads to *Y. enterocolitica* WA pYV-cured survival. Inhibition of autophagy by wild type *Y. enterocolitica* involves the T3SS effector YopE, and targeting of Rho-GTPases. Of note, Connor et al. (2015) propose that *Y. enterocolitica* 8081 pYV-cured is able to proliferate intracellularly within Raw264.7 macrophages, but the bacterial replication niche was not explored [41].

More recently, Valencia-Lopez and colleagues (2019) [42] explored infection of HeLa cells by the *Y. enterocolitica* WA pYV-cured strain previously used to investigate macrophage infection. In this work, upon correlative-light electron microscopy studies, the authors indicate that bacteria are found in YCVs characterized by autophagy-related, ultrastructural features, including the presence of double or multiple membranes. By exploring infection of wild type HeLa cells or with FIP200 deficiency (which affects classical canonical autophagy, but not LAP) through correlative light/electron microscopy (CLEM), the authors conclude that canonical autophagy is the main cellular process subverted by *Y. enterocolitica* (and not preferentially LAP, as proposed by Moreau et al. for *Y. pseudotuberculosis*). Fluorescence microscopy indicates that YCVs are decorated by LC3-GFP as well as by the LC3 adapter p62 and ubiquitin. Interestingly, and as previously observed for *Y. pseudotuberculosis*, Valencia-Lopez and co-authors demonstrate that *Y. enterocolitica* also escapes autolysosomal maturation by inhibiting fusion with lysosomes, resulting in bacterial-containing, non-acidic compartments devoid of proteolytic activity, in which bacteria replicate. Dead bacteria are not able to trigger LC3-GFP recruitment to the YCV, and they are found in acidic compartments suggesting that the induction of autophagy and acidification inhibition are active processes requiring metabolically active bacteria. As opposed to observations performed by Deuretzbacher et al. in macrophages, invasin signaling is not sufficient for autophagy induction or inhibition of lysosomal fusion. Interestingly, the authors show that half of the internalized *Y. enterocolitica* are present in phagosomes which do not engage autophagy and are LAMP-1 positive but devoid of LC3, indicating that this bacterial population is eliminated by a classical lysosomal pathway characterized by phagosomal acidification and proteolytic degradation [40]. The authors also suggest that autophagy could be subverted by *Y. enterocolitica* to promote a non-lytic egress from epithelial cells [42].

## 5. *Yersinia pestis*

Replication of wild type *Y. pestis*, or strains lacking pCD1, had been observed in mouse macrophages [15,43], and ultrastructural studies suggested that bacterial replication takes place within a phagolysosomal compartment [44]. Pujol and colleagues (2009) [45] determined that wild type *Y. pestis* KIM5+ multiplies efficiently in BMDMs, and that a subpopulation of bacteria could be found in double membrane compartments. Autophagic flux analyses demonstrated that conversion of LC3 I towards LC3 II is augmented upon BMDM infection by *Y. pestis*, and fluorescent microscopy identified recruitment of LC3-GFP to YCVs. As previously observed for enteropathogenic *Yersinia*, *Y. pestis*-containing vacuoles start as tight-fitting compartments, but progressively enlarge and become spacious vacuoles at late infection time points, which sustained bacterial replication. In addition, as previously observed for *Y. pseudotuberculosis*, *Y. pestis* avoids acidification of its compartment. However, differently from what had been reported for *Y. pseudotuberculosis*, *Y. pestis* survives equally well in BMDMs proficient (wild type) or deficient (ATG5 mutant) for autophagy, suggesting that *Y. pestis* may not require autophagy for its survival in macrophages [45].

Two recent studies have also implicated the small GTPases of the Rab family as key effectors allowing *Y. pestis* to avoid killing by macrophages. Rab1 is normally associated to ER-to-Golgi trafficking, but the Rab1b isoform has been specifically involved in autophagosome formation [46]. Connor and colleagues (2015) [41] first showed that siRNA inactivation of Rab1b significantly reduced the survival of *Y. pestis* CO92 pCD1-cured in RAW264.7 macrophages, while it does not affect bacterial entry. The authors then showed that Rab1b was required for *Y. pestis* to block YCVs acidification as well as fusion with LAMP-1 positive compartments. Fluorescent microscopy experiments showed that Rab1b was directly recruited to YCVs, but surprisingly Rab1b siRNA inactivation did not affect *Y. pestis* recruitment of LC3, suggesting that Rab1b controls a signaling cascade that is not directly related to autophagosome formation. A subsequent genome-wide siRNA screen performed by the same team (Connor et al. 2018) [47] showed that the small GTPases Rab4 and Rab11b are also recruited to the YCVs and contribute to *Y. pestis* survival in macrophages. Rab4a would cooperate together with Rab1b in the early steps of infection by inhibiting YCVs fusion with lysosomes, avoiding acidification. On the other hand, Rab11b, which remains sequestered by the YCVs, would allow disruption of the host cell endosomal recycling pathway, favoring bacterial replication at later infection time points (Figure 1) [47].

## 6. *Yersinia ruckeri*

Another species from the *Yersinia* genus, *Y. ruckeri*, is a fish pathogen responsible for enteric redmouth disease [48]. Experiments in fathead minnow-derived epithelial cells (FHM), in a liver cell line from the rainbow trout (R1), as well as in salmon kidney (ASK, SHK) and embryonic cell lines (CHSE-214) indicate that *Y. ruckeri* has the capacity to invade host cells [49,50,51]. However, the intracellular stage appears to be transient, leading to host cell death [51] or bacterial degradation [50], suggesting that *Y. ruckeri* is not able to multiply or survive inside epithelial cultured cells. Interestingly, Ryckaert and colleagues (2010) demonstrated by gentamicin protection assays that *Y. ruckeri* can invade and survive within rainbow trout macrophages [52]. Using transmission electron microscopy, *Y. ruckeri* is detected in infected macrophages within double membrane vacuoles, which display autophagosomal features. Intracellular bacteria replicate and survive within these compartments for at least 24 h after infection [52]. Nevertheless, potential autophagic markers of this YCV, its formation, and maturation process remain to be deciphered.

## 7. Discussion

*Y. enterocolitica*, *Y. pseudotuberculosis,* and *Y. pestis* produce potent effectors (Yops) that block internalization within host cells [9,10], and therefore these microorganisms have been traditionally considered as mainly extracellular pathogens. Indeed, in vivo, bacterial replication has been detected in the extracellular environment of infected organs [8,14]. However, bacterial intracellular stages have been shown to play a major role in vivo for early infection of the intestinal barrier, or intestinal persistence in the case of enteric yersinioses [3,4,5]. Concerning interaction with phagocytic cells, factors required for intracellular survival of *Y. pseudotuberculosis* in macrophages have been shown to be required for bacterial virulence in vivo [16]. Since these factors are shared with *Y. pestis* [53], it has been also proposed that for plague, bacterial intracellular survival and/or replication within phagocytic cells might be critical for early pathogen transport from the skin (upon flea bit) to draining lymph nodes [54]. Recent in vivo models of *Y. pestis* skin infection suggest that transport to draining lymph nodes might not require intracellular transport by macrophages, dendritic cells or neutrophils [55,56]. However, an intracellular phase of bacterial proliferation within phagocytic cells during late in vivo infections cannot be formally excluded for *Y. pestis* [57]. This conclusion also stands for *Y. enterocolitica* and *Y. pseudotuberculosis*.

Independent evidence generated with the three species *Y. enterocolitica*, *Y. pseudotuberculosis*, and *Y. pestis* in both phagocytic and epithelial cells clearly converges with the idea that pathogenic *Yersinia* activate and subvert autophagy to replicate and survive within host cells. Initial experiments from Deuretzbacher et al. (2009) [40] suggested that autophagy activation favored *Y. enterocolitica* killing by macrophages. As discussed by Valencia-Lopez and colleagues (2019) [42], the use of tetracycline-inactivated bacteria in the former study probably influenced bacterial destruction, proposing therefore that metabolically active *Y. enterocolitica* may survive within professional phagocytes, as observed for *Y. pseudotuberculosis* and *Y. pestis.* As mentioned, *Y. ruckeri* has been also described to persist in fish macrophages within double membrane-bound compartments [52], suggesting that autophagy subversion could also be present in this *Yersinia* species.

The molecular effectors that allow pathogenic *Yersinia* to subvert autophagy have not been identified. The work of Valencia-Lopez et al. [42] indicates that active bacterial metabolism is required for macroautophagy subversion by *Y. enterocolitica*. The plasmidic T3SS is dispensable, as shown by experiments in which pYV- or pCD1-cured bacteria are able to induce the formation of spacious autophagosomes. Interestingly, some *Y. enterocolitica* strains encode a chromosomal T3SS named Ysa (*Yersinia* secretion apparatus) that is required for bacterial intracellular growth in *Drosophila* S2 cells [58]. This system is also present in *Y. ruckeri* [59], but its potential contribution to intracellular vacuolar remodelling remains to be demonstrated. The chromosomally-encoded invasin does not seem sufficient to promote autophagy subversion, and the inactivation of inv in *Y. pestis* also argues against a functional role of this surface protein in promoting a phenotype that seem shared by the three bacterial species discussed in this article. Valencia-Lopez et al. (2019) [42] suggest that phospholipases, produced by several *Yersinia* species, might trigger autophagy by damaging the membrane of the YCVs and exposing bacteria to the cytosol, allowing therefore the recruitment of the phagophore formation machinery. Differences observed between the specific autophagy cascades triggered by pathogenic *Yersinia* (LAP in the case of *Y. pseudotuberculosis*, canonical autophagy in the case of *Y. enterocolitica* during epithelial cell invasion) could be explained by variations in the various sets of chromosomal effectors encoded by each bacterial species which would mediate autophagy induction.

Several additional observations from the work of Valencia-Lopez and colleagues (2019) deserve attention [42]. First, they showed that different bacterial subpopulations may coexist within host cells: one of these subpopulations will successfully multiply upon autophagy subversion, but the other subpopulation will be subjected to lysosomal elimination. The effectors involved at this molecular crossroad are not known, and could involve Rab GTPases on the host side. Second, their results indicate that autophagy is also required for non-lytic egress of *Y. enterocolitica* from infected cells. They draw attention to the fact that Rab11b, VAMP3, and VAMP7, which have been associated to the formation of autophagosomes by *Y. pestis* and *Y. pseudotuberculosis*, are regulatory elements of the exocytic machinery. It would therefore be interesting to investigate the potential contribution of a general autophagy-related mechanism involved in promoting bacterial escape from host cells.

Pathogenic *Yersinia* species have been instrumental models to explore the molecular interaction between pathogens and host cells. The issues raised by the study of autophagy subversion by *Y. enterocolitica*, *Y. pseudotuberculosis*, and *Y. pestis* (and also potentially *Y. ruckeri*) reviewed in here, clearly indicate that these bacteria have still many secrets to unfold.

## Figures and Tables

**Figure 1 biomolecules-10-01637-f001:**
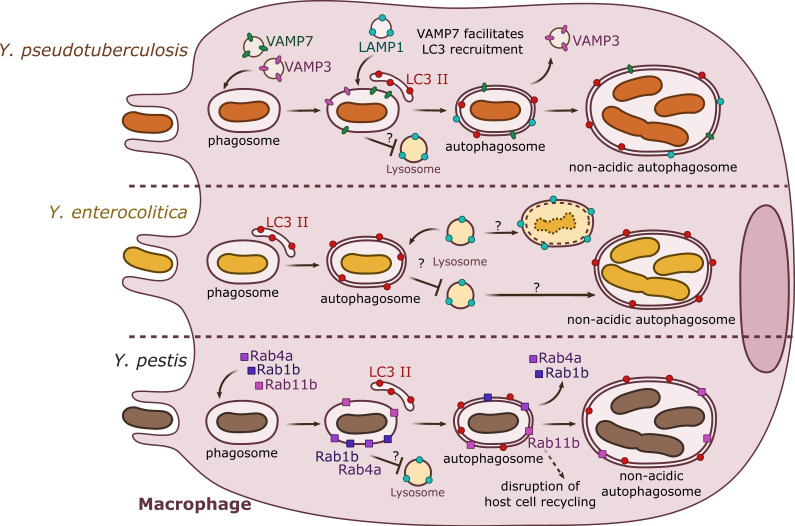
Summary results from independent studies of autophagy induction by *Y. pseudotuberculosis, Y. enterocolitica*, and *Y. pestis* in macrophages. *Y. pseudotuberculosis* (top) recruits VAMP3 and VAMP7 to the YCV early in the invasion process. VAMP7 facilitates the recruitment of LC3-autophagic membranes, forming an autophagosome with double or multiple membranes. Rab1b (not represented) is important for bacterial survival. *Y. pseudotuberculosis* survives and replicates in a non-acidic autophagosome. *Y. enterocolitica* (center) is present in a double- or multiple-membrane autophagosomal compartment positive for LC3. Depending on the strain and its pathogenicity, *Y. enterocolitica* seems to survive in autophagosomes. *Y. pestis* (bottom) targets Rab GTPases (1b, 4a, 11b) to the phagosome. Rab1b and Rab4a participate in the inhibition of acidification and thus are involved in bacterial survival. Rab11b is sequestered to the autophagosome over the course of infection, which leads to a global inhibition of host endosomal recycling. *Y. pestis* proliferates in a non-acidic autophagosome.

**Figure 2 biomolecules-10-01637-f002:**
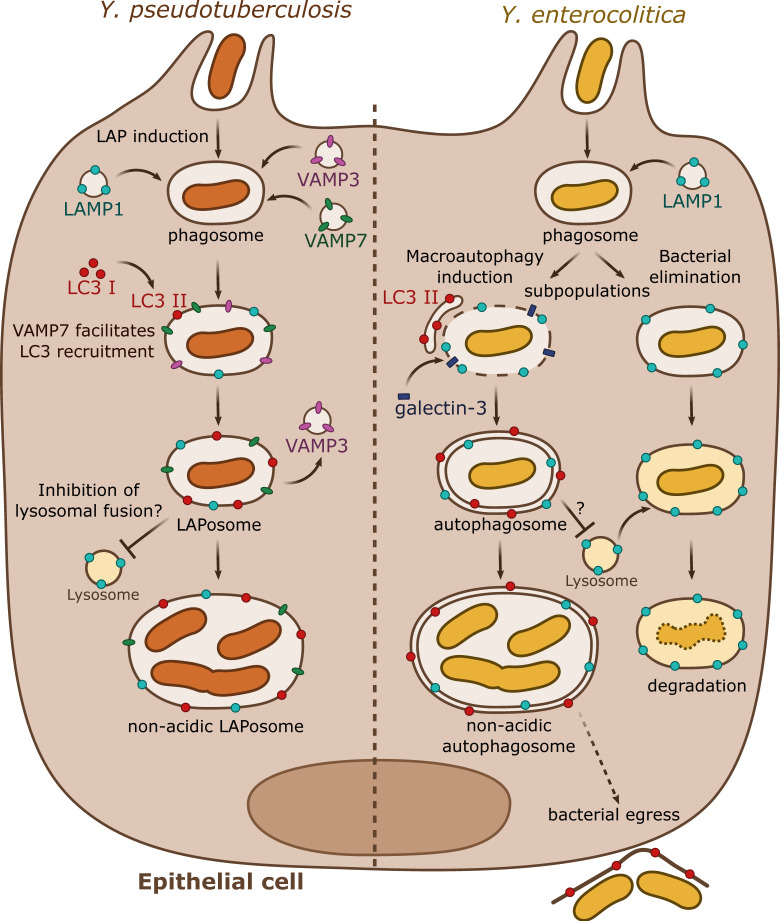
Summary results from independent studies of autophagy subversion by enteropathogenic *Yersinia* in epithelial cells. *Y. pseudotuberculosis* (left) activates the LAP autophagic pathway during epithelial infection. VAMP3 and VAMP7 are sequentially targeted to the phagosome. VAMP3 favors the commitment towards a single-membrane compartment. VAMP7 participates to the recruitment of LC3-II directly to the phagosome forming the LAPosome. *Y. pseudotuberculosis* survives and multiplies within a non-acidic single membrane LAPosome. *Y. enterocolitica* (right) can survive or be degraded in epithelial cells. A subpopulation follows the lysosomal degradative pathway whereas the rest activates the macroautophagy pathway by recruiting LC3-positive autophagic membranes. Galectin-3, a marker of damaged endomembrane, is also recruited to some YCVs, suggesting their potential disruption. *Y. enterocolitica* survives and replicates in a non-acidic double or multiple membrane autophagosome blocked in its maturation process. The autophagosome seems to support bacterial egress without cell lysis.

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
