# Peer review of "Autophagy and Intracellular Membrane Trafficking Subversion by Pathogenic Yersinia Species"

_biomolecules, 2020, doi:10.3390/biom10121637_

Round 1

Reviewer 1 Report

The review article by M. Lemarignier and J. Pizarro-Cerda compiles the relevant findings of a number of studies that had addressed the role of autophagy in the biology of Yersinia-infected host cells. All three pathogenic Yersinia species Y. pestis, Y. pseudotuberculosis and Y. enterocolitica were described to trigger autophagy and to survive and multiply in infected cells. Potential similarities and overlaps in these features by the diverse pathogenic Yersinia species, however, have not yet been specified in a specific review article. The present article now fills this gap and summarizes and discusses common and differing features in the implication of autophagy in determining the intracellular fate of the diverse Yersinia species. The article provides a balanced and adequate interesting short survey of the major results and conclusions of the original research work to this issue. It is highlighted that autophagy induction may have evolved as strategy for survival and replication in phagocytic and epithelial cells for all three pathogenic Yersinia species.

There are some minor comments that may additionally help to improve the article:

  1. In the legend to Fig.1 it is stated that Y. pseudotuberculosis recruits Rab1b which may play role in inhibition of YCV acidification. The cited reference, however, has shown this coherence for Y. pestis, but not for Y. pseudotuberculosis. It would therefore be desirable that either an alternative adequate reference is provided or it is stated that this conclusion is deduced from the studies on Y. pestis.
  2. In the same manner, it is stated that Rab11b is sequestered to the autophagosome in case of Y. pestis infection (legend to Fig. 1). However, this has apparently not been formally shown by the cited references. It should therefore be specified whether this statement is speculative or whether it is deduced from a specific reference that should be mentioned.
  3. It appears more precise to state in the legend to Fig. 2 that Galectin-3 recruitment to YCVs suggests vacuole membrane disruption than degradation.
  4. “Y. enterocolitica” on page 5, line 205, should be in italics
  5. Page 6, line 226: Figure 1 instead of Figure 2 should be indicated here.

Author Response

We thank the Reviewer 1 for the very useful suggestions.

Please find below our answers, point by point.

REVIEWER 1

 Point 1: In the legend to Fig.1 it is stated that Y. pseudotuberculosis recruits Rab1b which may play role in inhibition of YCV acidification. The cited reference, however, has shown this coherence for Y. pestis, but not for Y. pseudotuberculosis. It would therefore be desirable that either an alternative adequate reference is provided or it is stated that this conclusion is deduced from the studies on Y. pestis.

Answer: We thank the reviewer for this comment. Indeed, the reference mentioned in the text (Connor et al. 2015) only makes direct reference to Y. pestis (the reference to Y. pseudotuberculosis in Fig. S3 is only indirect). Accordingly, and in agreement with the comment from the reviewer, we have removed Rab1 from the upper panel of Fig. 1, and a comment has been made in line 194 of the revised version of our manuscript, indicating that 'Rab1b (not represented) is important for baterial survival', making reference to the Fig. S3 of Connor et al. 2015).

Point 2: In the same manner, it is stated that Rab11b is sequestered to the autophagosome in case of Y. pestis infection (legend to Fig. 1). However, this has apparently not been formally shown by the cited references. It should therefore be specified whether this statement is speculative or whether it is deduced from a specific reference that should be mentioned.

Answer: In the reference Connor et al. 2018 (mBio), the authors illustrate in Figure 5B the recruitment of Rab11b to the Yersinia pestis-containing vacuole, in Figure 5E they quantify this significant early recruitment, and in Figure 5H they show how Rab11b remains associated to the Yersinia pestis-containing vacuole. These results therefore support the scheme we propose in Figure 1. For clarity, we have added a phrase in the main text in which the sequestering of Rab11b to the Yersinia pestis-containing vacuole is associated to the reference Connor et al. 2018 (line 313 of the revised version of our manuscript): 'On the other hand Rab11, which remains sequestered by the YCVs...'.

Point 3: It appears more precise to state in the legend to Fig. 2 that Galectin-3 recruitment to YCVs suggests vacuole membrane disruption than degradation.

Answer:  We have changed the legend of Fig. 2 according to the suggestion of the reviewer (line 242 of the revised version of our manuscript).

Point 4: “Y. enterocolitica” on page 5, line 205, should be in italics

Answer: We have changed the text according to the comment of the reviewer, and we have modified Y. enterocolitica and Yersinia pseudotuberculosis as 'enteropathogenic Yersinia' (line 293 of the revised version of our manuscript).

Point 5: Page 6, line 226: Figure 1 instead of Figure 2 should be indicated here.

Answer: We have indicated Figure 1 in the position suggested by the reviewer. We thank the reviewer for the suggestion (line 320 of the revised version of our manuscript).

Reviewer 2 Report

Overview:

The present manuscript constitutes review on the mechanisms through which members of the Yersinia genus subvert the host’s mechanisms of autophagy and membrane trafficking to enhance their infectious abilities. Yersinia is a major genus of pathogenic bacteria in both humans and other animals and therefore, this review should be of interest to the readers. Overall, I find the manuscript of very good quality, if a bit short. I have a few minor comments, detailed below, but the article remains of excellent quality. Therefore, I recommend its publication following minor revisions.

Minor comments:

L 34: In fact, two different families of T3SS are represented in the genus Yersinia with some species harbouring both types of T3SS alongside each other.

L 48: Please replace “highlighting that” with “describing how”.

L 52: I would suggest expanding on autophagy a little bit more: evolutionary benefits and the different steps in the process.

L 105: ”harbours” instead of “is decorated by the”.

L 164: Replace the sentence with: “These authors observed intracellular bacteria in multi-membrane compartments using electron microscopy as well as bacteria associated to LC3-GFP-positive vacuoles, observed using fluorescence microscopy”.

L 226: I would suggest adding the description of other members of the Yersinia genus, notably Y. ruckeri as this is a somewhat unique member of the genus.

L 254: More detail should be given about the findings of Valencia-Lopez et al.

Author Response

We thank the Reviewer 2 for the very useful suggestions.

Please find below our answers, point by point.

REVIEWER 2

Point 1 (L 34): In fact, two different families of T3SS are represented in the genus Yersinia with some species harbouring both types of T3SS alongside each other.

Answer: We thank the reviewer for this comment. Indeed, some Yersinia enterocolitica display two additional chromosomal T3SS: the Ysa system that injects Yersinia secreted proteins (Ysps), and the flagellar export apparatus that allows secretion of the phospholipase YplA. Both systems are involved in Y. enterocolitica virulence, but their specific function has remained obscure. A paper from the group of Virginia Miller in 2013 suggests that the Ysa system participates to the intracellular life of Y. enterocolitica, using a Drosophila S2 cellular system. In the revised version of our manuscript (lines 368 to 371), we now mention in the discussion the reference to the the Ysa system, which also exists in Y. ruckeri.

Point 2 (L 48): Please replace “highlighting that” with “describing how”.

Answer: We have replaced the text as suggested by the reviewer (line 64 in the revised version of our manuscript).

Point 3 (L 52): I would suggest expanding on autophagy a little bit more: evolutionary benefits and the different steps in the process.

Answer: As suggested by the reviewer, we have expanded the autophagy section, providing more detail on the different steps in the process, and mentioning evolutionary benefits (lines 68 to 79, as well as lines 91 to 94 of the revised version of our manuscript).

Point 4 (L 105): ”harbours” instead of “is decorated by the”.

Answer: We have changed the text as suggested by the reviewer (line 154 of the revised version of our manuscript).

 Point 5 (L 164): Replace the sentence with: “These authors observed intracellular bacteria in multi-membrane compartments using electron microscopy as well as bacteria associated to LC3-GFP-positive vacuoles, observed using fluorescence microscopy”.

Answer: We thank the reviewer for the suggestion and we have changed the text accordingly (lines 249 to 251 of the revised version of our manuscript).

Point 6 (L 226): I would suggest adding the description of other members of the Yersinia genus, notably Y. ruckeri as this is a somewhat unique member of the genus.

Answer: As suggested by the reviewer, we have added a paragraph in which we summarize the reports of interactions of Y. ruckeri with host cells (lines 317 to 330 of the revised version of our manuscript).

Point 7 (L 254): More detail should be given about the findings of Valencia-Lopez et al.

Answer: In the new version of our manuscript, we have provided more details about the findings from the work of Valencia-Lopez et al., as suggested by the reviewer (lines 274 to 275, 365 to 366, and 382 to 386 of the revised version of our manuscript).

Reviewer 3 Report

See attached file

Author Response

We thank the Reviewer 3 for the very useful suggestions.

Please find below our answers, point by point.

REVIEWER 3

Point 1: Information and discussion about the existence and role of intracellular replication during in vivo infection will strengthen the motivation of this review and relevance of the topic. This should be included at the end of the introduction and in the discussion. I am convinced such an overview and discussion would be of great interest for the readers.

Answer: We thank the reviewer for the comment. We fully agree with the reviewer's opinion and in our original introduction, references to the work of Pepe et al. 1993, Clark et al. 1998 and Fahlgren et al. 2014 were precisely selected to make already the point to the importance of bacterial intracellular life in in vivo infection models. Following the reviewer's suggestion, we have detailed in the introduction some of the relevant aspects of these already cited references (lines 40 to 43 of the revised version of our manuscript), and in the discussion (lines 332 to 353 of the revised version of our manuscript), we have expanded these notions, adding additional references.

Point 2: The figures are nicely drawn, but the figure legend titles should be changed so they describe what the figure indeed shows, i.e. summary results from independent studies of autophagy induction by Yersinia. This is important since they illustrate data that are published separately and therefore not necessarily subjected to exactly the same approaches. Although pointing to a similar mechanism, the details of the resulting concluding models will differ for each study. This is nicely concluded in the discussion, where, for example, it is mentioned that Y. enterocolitica also can be expected to survive within professional phagocytes (line 234), similar to the other Yersinia species.

Answer: We have modified the figure legends following the suggestion of the reviewer (lines 190 to 191, and 216 to 217 of the revised version of our manuscript).

Point 3: Some sentences appear complicated and there are many too long sentences (ex line 38-41), so it is required to go through the text and change accordingly.

Answer: We thank the reviewer for the suggestion. We have shortened the long sentence in lines 38-41 (now lines 37 to 40 of the revised version of our manuscript), and we have also modified other long sentences through the text, as suggested (lines 33 to 36 of the revised version of our manuscript for example).

Point 4: Line 49, “favor a bacterial replication”

Answer: We have changed the text as suggested by the reviewer (line 65 of the revised version of our manuscript).

Point 5: The very long sentence line 142-145 needs to be both shortened and clarified, the wording “participates to” is strange Same on line 219, participates to?

Answer: We have shortened the phrase indicated by the reviewer (lines 210 to 214 of the revised version of our manuscript). We also replaced 'participates' by 'controls' in the two phrases highlighted by the reviewer (line 212 and line 311 of the revised version of our manuscript).

Point 6: Line 177 take out the word “the”

Answer: We have corrected the text following the suggestion from the reviewer (line 264 of the revised version of our manuscript).